# Conformational plasticity and allosteric communication networks explain Shelterin protein TPP1 binding to human telomerase

Simone Aureli [1,2,3,4], Vince Bart Cardenas[1,4], Stefano Raniolo [1,4] & Vittorio Limongelli [1✉]

The Shelterin complex protein TPP1 interacts with human telomerase (TERT) by means of the TEL-patch region, controlling telomere homeostasis. Aberrations in the TPP1-TERT hetero-dimer formation might lead to short telomeres and severe diseases like dyskeratosis congenita and Hoyeraal-Hreidarsson syndrome. In the present study, we provide a thorough characterization of the structural properties of the TPP1's OB-domain by combining data coming from microsecond-long molecular dynamics calculations, time-series analyses, and graph-based networks. Our results show that the TEL-patch conformational freedom is influenced by a network of long-range amino acid communications that together determine the proper TPP1-TERT binding. Furthermore, we reveal that in TPP1 pathological variants Glu169Δ, Lys170Δ and Leu95Gln, the TEL-patch plasticity is reduced, affecting the correct binding to TERT and, in turn, telomere processivity, which eventually leads to accelerated aging of affected cells. Our study provides a structural basis for the design of TPP1-targeting ligands with therapeutic potential against cancer and telomeropathies.

[1] Faculty of Biomedical Sciences, Euler Institute, Università della Svizzera italiana (USI), via G. Buffi 13, Lugano CH-6900, Switzerland. [2]Present address: Institute of Pharmaceutical Sciences of Western Switzerland, University of Geneve, Rue Michel-Servet 1, Geneva CH-1211, Switzerland. [3]Present address: Swiss Institute of Bioinformatics, University of Geneve, Geneva CH-1206, Switzerland. [4]These authors contributed equally: Simone Aureli, Vince Bart Cardenas, Stefano Raniolo. ✉email: vittoriolimongelli@gmail.com

Telomeres are an ensemble of proteins, noncoding DNA, and RNA that protect chromosomes' termini from unwanted events like recombination and degradation, thus being crucial for cell lifespan (Fig. 1a)[1–5]. While in normal somatic cells telomeric DNA progressively shortens with each round of cell division leading to cell senescence, in cancer cells, the telomeric DNA length is preserved, allowing the tumor to continue proliferating. In particular, telomere lengthening is ensured in about 85% of cancer cells by the overexpression of the telomerase enzyme[6,7], which adds hexanucleotidic sequences to the 3' ends of chromosomes[8]. Telomerase is a ribonucleoprotein complex consisting of a protein subunit (TERT) that works as a reverse transcriptase using a specific RNA component (TR) as the template (Fig. 1b). Four domains characterize TERT, three of which form the conserved ring-shaped catalytic core and are (1) the "Telomeric RNA Binding Domain" (TRBD); (2) the "Reverse Transcriptase" (RT); and (3) the "C-Terminal Extension" (CTE)[9]. The fourth domain is the "Telomerase Essential N-terminal" (TEN) domain, deputed to enhance telomerase processivity and its recruitment to telomeres[10–13]. During its action, TERT is assisted by several regulatory proteins, among these being the Shelterin complex protein TPP1 playing a key role (Fig. 1c)[14–17]. In fact, TERT associates with telomeres by interacting with the "Oligosaccharide/Oligonucleotide-binding" domain (OB-domain) of TPP1[6,15,18,19], forming a binary complex crucial for telomere processivity. The region of the OB-domain responsible for binding to telomerase is named "TPP1 E and L rich-patch" (TEL-patch) due to the abundance of glutamate and leucine residues in this region (Glu168, Glu169, Glu171, Leu183, Leu212 and Glu215)[18]. Indeed, mutation or deletion of residues in the TEL-patch, or close to this region, could lead to pathologies known as Telomeropathies, characterized by a fast shortening of the telomeres or damage of DNA. Among them are Dyskeratosis Congenita (DC) and its more severe variant named Hoyeraal-Hreidarsson syndrome (HHS). The former might be induced by the single-point mutation Leu95Gln on the TPP1's OB-domain, whereas the latter is due to the deletions of either Glu169 or Lys170 at the TEL-patch (Glu169Δ and Lys170Δ, respectively).[20–23]. Patients affected by these disorders exhibit growth retardation, microcephaly, cerebellar hypoplasia, immune deficiency, aplastic anemia, and bone marrow failure with poor life expectancy[20,21].

In this scenario, the understanding of the molecular basis of DC and HHS is the first, paramount step toward the rational development of efficacious treatments. This requires the elucidation of the effect of Glu169Δ, Lys170Δ, and Leu95Gln mutations in TPP1 on the formation of the binary complex with TERT. The recently disclosed cryo-EM structures of TPP1/TERT structures[24,25] have prompted us to investigate the functional conformational changes of TPP1 in its wild-type form (WT), as well as in the pathological variants, and how these impact the binding with TERT. In particular, we have investigated the structural properties of the TPP1 OB-domain using state-of-the-art computational techniques and the available experimental data. First, we elucidated the dynamical and structural properties of WT TPP1 by means of extensive atomistic molecular dynamics (MD) simulations. Then, MD simulations carried out on the DC-prone Leu95Gln mutant and the HHS-prone Glu169Δ and Lys170Δ variants of the OB-domain were compared with those on the WT form through time-series data analysis, including principal component analysis (PCA), cross-correlation analysis (CCA) and graph-based structure network analysis. Such investigations revealed the presence of long-range communication networks between different regions of the OB-domain and the TEL-patch. An alteration of such allosteric networks leads to a decreased binding affinity of TPP1 towards TERT, with the consequent inhibition of telomere processivity. Evidence of this phenomenon is given by the investigations performed on the Glu169Δ, Lys170Δ, and Leu95Gln phenotypes. In fact, although such mutations are located in different regions of the OB-domain, they all affect TPP1's conformational plasticity, weakening its binding interaction with TERT's hTEN, as shown by both protein-protein docking simulations and μs-long MD calculations performed on the heterodimeric complex. An explanatory movie of the effect of Lys170Δ on TPP1 functional dynamics is available in the "Data Availability" section (see "Supplementary Video 1"). The PDB structures of the wild type and mutant forms of TPP1 both as monomer and in complex with TERT are released as "Supplementary Data 1". Such structures, alongside the interaction network between amino acids of WT TPP1 and its pathological variants, offer a molecular rationale and solid basis for the understanding of the functional mechanism of TPP1 and the Shelterin complex in general. Such wealth of data paves the way to structure-based drug discovery campaigns where compounds designed to work as allosteric enhancers of the TPP1-TERT binding interaction might contrast telomeropathies like HHS, while compounds acting as allosteric inhibitors of the TPP1-TERT binary complex might represent a novel class of anticancer agents.

## Results and discussions

The scope of our study is to characterize the structural properties of the shelterin protein TPP1 under physiological and pathological conditions (i.e., HHS Glu169Δ and Lys170Δ variants and DC Leu95Gln mutant). To this end, we carried out a total of 12.0

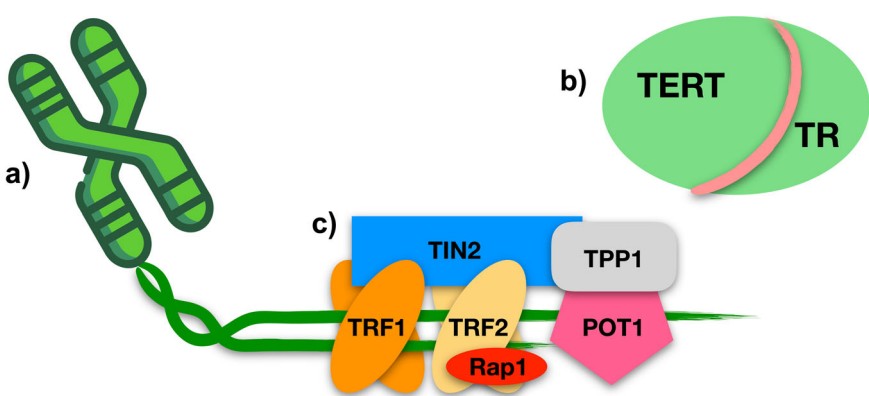

**Fig. 1 Telomeres and the telomeres' proteome. a** Telomeres are portions of noncoding DNA located at the chromosome termini. **b** The Telomerase ribonucleoprotein, composed of the enzyme TERT (green) and the telomeric RNA TR (pink). **c** The 6-membered Shelterin complex, composed of the double-strand binding proteins RAP1 (red), TRF2 (yellow), TRF1 (orange), and TIN2 (dodger-blue), together with the single-strand binding proteins TPP1 (gray) and POT1 (magenta).

 

μs atomistic MD calculations on the WT, Glu169Δ, Lys170Δ, and Leu95Gln TPP1 OB-domain (Fig. 2a), and performed a number of time-series analyses on the simulated data. The results are discussed in the following paragraphs.

**Structural properties of WT, Glu169Δ, Lys170Δ and Leu95Gln TPP1.** We performed three replicas, each 1.0 μs long, of classical MD calculations for every TPP1 variant. The data obtained from the replicas were collected and analyzed as follows. We first computed the Root Mean Square Deviation (RMSD) of the backbone atoms of the secondary structures during the simulations in order to inspect the overall conformational behavior of the protein. In Fig. 2b, the low average RMSD values (~1.0 Å) computed for the secondary structure Cα atoms as a function

of simulation time indicate good conformational stability for all systems. This result is further confirmed by the cluster analysis of the conformational states visited by the proteins during the MD simulations that led to a very small number of cluster families (see Supplementary Table S1, Supplementary Note 1, and the "Methods" section for details). However, few differences between WT, Glu169Δ, Lys170Δ, and Leu95Gln TPP1 arise at the level of the TEL-patch region. In this regard, in WT TPP1 the three most populated conformation families, namely W1, W2, and W3 (Fig. 2c), present a different state of the TEL-patch. In particular, W1 shows a closed conformation, W2 an open conformation, while W3 a semi-open conformation. This finding reveals a certain conformational freedom of the TEL-patch that might be functional for the binding to hTEN. The PDB structures of W1, W2, and W3 are released as Supplementary Data 1.

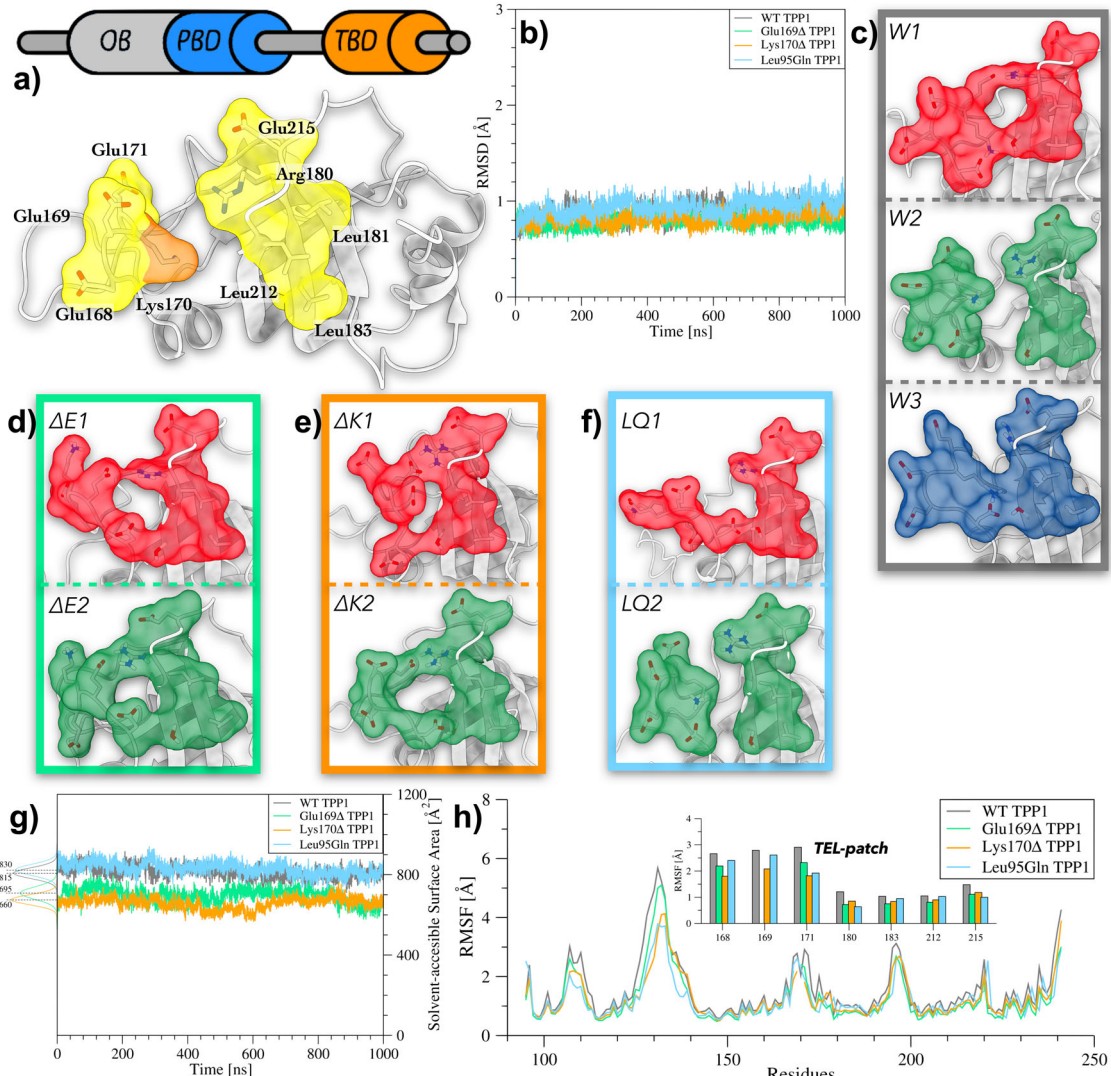

**Fig. 2 Comparison between structural features of WT TPP1, Glu169Δ TPP1, Lys170Δ TPP1, and Leu95Gln TPP1. a** Topology of TPP1 and the 3D structure of the OB-domain (gray). TEL-patch and Lys170 are represented through their solvent-accessible surface (yellow and orange, respectively). **b** Plot of the RMSD as a function of simulation time computed for the secondary structure Cα atoms of WT, Glu169Δ, Lys170Δ, and Leu95Gln TPP1 (gray, green, orange, and cyan, respectively). **c** Conformations assumed by the TEL-patch in cluster families W1, W2, and W3. The solvent-accessible surface is shown in red, green and blue, respectively. **d** Conformations assumed by the TEL-patch in cluster families ΔE1 and ΔE2. The solvent-accessible surface is shown in red and green. **e** Conformations assumed by the TEL-patch in cluster families ΔK1 and ΔK2. The solvent-accessible surface is shown in red and green. **f** Conformations assumed by the TEL-patch in cluster families LQ1 and LQ2. The solvent-accessible surface is shown in red and green. **g** Plot of the Solvent Accessible Surface Area (SASA) as a function of simulation time calculated for the TEL-patch in WT, Glu169Δ, Lys170Δ, and Leu95Gln TPP1 (gray, green, orange, and cyan respectively). **h** Plot of RMSF values computed for each residue of WT, Glu169Δ, Lys170Δ, and Leu95Gln TPP1 (gray, green, orange, and cyan respectively). The TEL-patch's residues are represented as histograms in the insets.

At variance with the WT, Glu169Δ TPP1 and Lys170Δ TPP1 show a remarkably decreased conformational freedom in the TEL-patch. Regarding Glu169Δ TPP1, only two conformational states were obtained from the cluster analysis, namely ΔE1 and ΔE2 (Fig. 2d). Both present the TEL-patch in the *closed conformation*. The structural analysis of these two states during the evolution of the MD simulation reveals the presence of two long-lasting interactions formed by Glu168 with Arg180, and by Asp166 with Ser210 (see Supplementary Fig. S1). A similar scenario was found in Lys170Δ TPP1, for which two main conformational states were identified through the cluster analysis and named ΔK1 and ΔK2 (Fig. 2e). The analysis of their structures revealed the presence of salt bridges between the TEL-patch residues that favor the closed conformation. In particular, the ΔK1 state is stabilized by the Glu169-Arg180 interaction, while the salt bridge Glu171-Arg180 characterizes ΔK2 (see Supplementary Note 1 and Supplementary Fig. S1). Interestingly, we found out that in both variants, the deletion of a charged residue—despite opposite in charge—induces the formation of strong long-lasting interactions able to stabilize the TEL-patch in a closed conformation, thus significantly reducing the TPP1 plasticity. As in the case of Glu169Δ TPP1, two conformational states were also found for the Leu95Gln mutant, i.e., LQ1 and LQ2 (see Fig. 2f ). Interestingly, the TEL-patch of LQ1 and LQ2 assumes a conformation similar to that of the WT in the semi-open W3 and open conformation W2, respectively. Upon closer inspection, it emerges that the mutated residue Gln95 engages a number of intraprotein H-bonds with residues such as Trp98, Arg113, Ala114, Asp148, and Gln216, which decreases the protein conformational freedom (Supplementary Note 2 and Supplementary Fig. S2). Such interactions cannot be formed in the WT by the apolar Leu95, with the final result that the WT has a higher plasticity, passing from the open to the semi-open and closed state. The PDB structures of ΔE1, ΔE2, ΔK1, ΔK2, LQ1, and LQ2 are released as Supplementary Data 1.

Prompted by this finding, we decided to investigate more deeply how these microscopic differences—at atomistic scale—could affect the macroscopic properties of TPP1. Therefore, we computed the solvent accessible surface area (SASA) of the TEL-patch and the root mean square fluctuation (RMSF) of each residue of WT, Glu169Δ, Lys170Δ, and Leu95Gln TPP1 during the MD simulations. In line with what we previously found, both Glu169Δ TPP1 and Lys170Δ TPP1 display lower SASA values than WT TPP1 (695–660 Å² vs 815 Å²), indicating a more compact state of the variants (Fig. 2g). On the other hand, as expected by the presence of the open and semi-open state, the Leu95Gln mutant shows SASA values comparable to that of WT (830 Å² vs 815 Å²). However, all the variants have RMSF values lower than those of the WT, especially in the TEL-patch, confirming the reduced conformational flexibility of TPP1 in these forms (Fig. 2h). Interestingly, our results show that the effect of the deletions/mutation is not limited to this region. In fact, all the variants provoke long-range allosteric effects in TPP1, reducing the conformational flexibility of regions even distant from TEL-patch, including the regions from Ser106 to Gln116 and from Asp123 to Gly141 (see Fig. 2h). This observation led us to perform a characterization of the TPP1 functional dynamics that is discussed in the following section.

**Effect of Glu169Δ, Lys170Δ, and Leu95Gln on TPP1 functional dynamics**. In order to elucidate the effect of the Glu169 and Lys170 deletions and Leu95Gln mutant on the functional dynamics of TPP1, we performed a principal component analysis (PCA) on the Cα atoms of TPP1 in the WT and the variant forms. Such a method allows identifying the key components (i.e.,

atoms) of a system that are responsible for large-scale motion endowed with a relatively long timescale[26]. In our case, using such an approach, we could detect the most relevant slow motion of the systems along the MD simulations and finally provide a graphical representation of the main conformational rearrangements in the three variants of TPP1 (Fig. 3). Specifically, we focused our analysis on three regions of TPP1 defined as follows:

- "Asp123-Gly141" (**I**), the longest loop of the TPP1 OB-domain structure (red in Fig. 3a);
- "Thr158-Glu178" (**II**), hosting the TEL-patch "Knuckle" motif, proposed by Grill et al.[22], where Glu168, Glu169, Lys170 and Glu171 are located (green in Fig. 3a);
- "Asp207-Val221" (**III**), where additional two residues belonging to the TEL-patch "Barrel part", namely Leu212 and Glu215, are located (blue in Fig. 3a).

In Fig. 3b–e, we report the projection of the module of the first eigenvector onto the protein structures for WT, Glu169Δ, Lys170Δ, and Leu95Gln TPP1, respectively. This represents the slowest timescale and largest conformational change in the protein structure. It is worth noting that a significant motion of **II** only occurs in WT TPP1, while a more rigid structure characterizes the three variants. A similar behavior is also found for **I**, which is endowed with reduced flexibility in Glu169Δ, Lys170Δ, and Leu95Gln TPP1. Further details about the PCA analyses are reported in Supplementary Note 3, Supplementary Table S2, and Supplementary Figs. S3–S6.

Along with PCA, we performed a cross-correlation analysis (CCA), which allows identifying short- and long-range allosteric effects between residues[26]. In such a way, we could elucidate communication networks in the OB-domains. Regarding WT TPP1, protein regions **I**, **II** and **III** show anti-correlated motions —i.e., negative Pearson coefficients—each with the other (Fig. 3f ). On the other hand, these signals are lost in the OB-domain variants (Fig. 3g–i). This analysis confirms the presence of a concerted motion in the WT between the TEL-patch region and the rest of the protein, which is suppressed by the deletions in the TEL-patch—as also suggested in a previous study[22]—or by the Leu95Gln mutation.

A complementary result was obtained by calculating the Protein Structure Networks (PSNs) for both WT TPP1 and its variants, using a graph-based approach that assesses the strength of the inter-residue interactions along an MD trajectory[27] (see Supplementary Note 4). In detail, the PSN of WT TPP1 is characterized by a higher number of edges with respect to the PSN of the variants (see Supplementary Fig. S7A–D). A distinguishing feature of Glu169Δ TPP1 and Lys170Δ TPP1 is the dense communication network between the regions **II** and **III** of the TPP1's OB-domain, in agreement with the finding that TEL-patch assumes a more compact closed conformation in such forms. At variance with Glu169Δ and Lys170Δ, Leu95Gln TPP1 shows few edges between regions **II** and **III**, while a strong communication network is established between the N-terminus (where Gln95 is located) and region **III**. This outcome is in line with the previously reported observation that Leu95Gln TPP1 can assume an open and semi-open conformation, while it is not able to assume a closed conformation due to the intraprotein contacts established by Gln95. Regarding the rest of the OB-domain, overall all the variants have a lower number of intraprotein connections with respect to the WT (see Supplementary Fig. S8A–D), expression of a reduced protein plasticity.

Taken together, our findings indicate that WT TPP1's conformational plasticity is regulated by a concerted motion between the regions **I**, **II** and **III**, and the deletions/mutant inhibit it through two molecular mechanisms. The first one is due to the formation of long-lasting interactions between residues at region

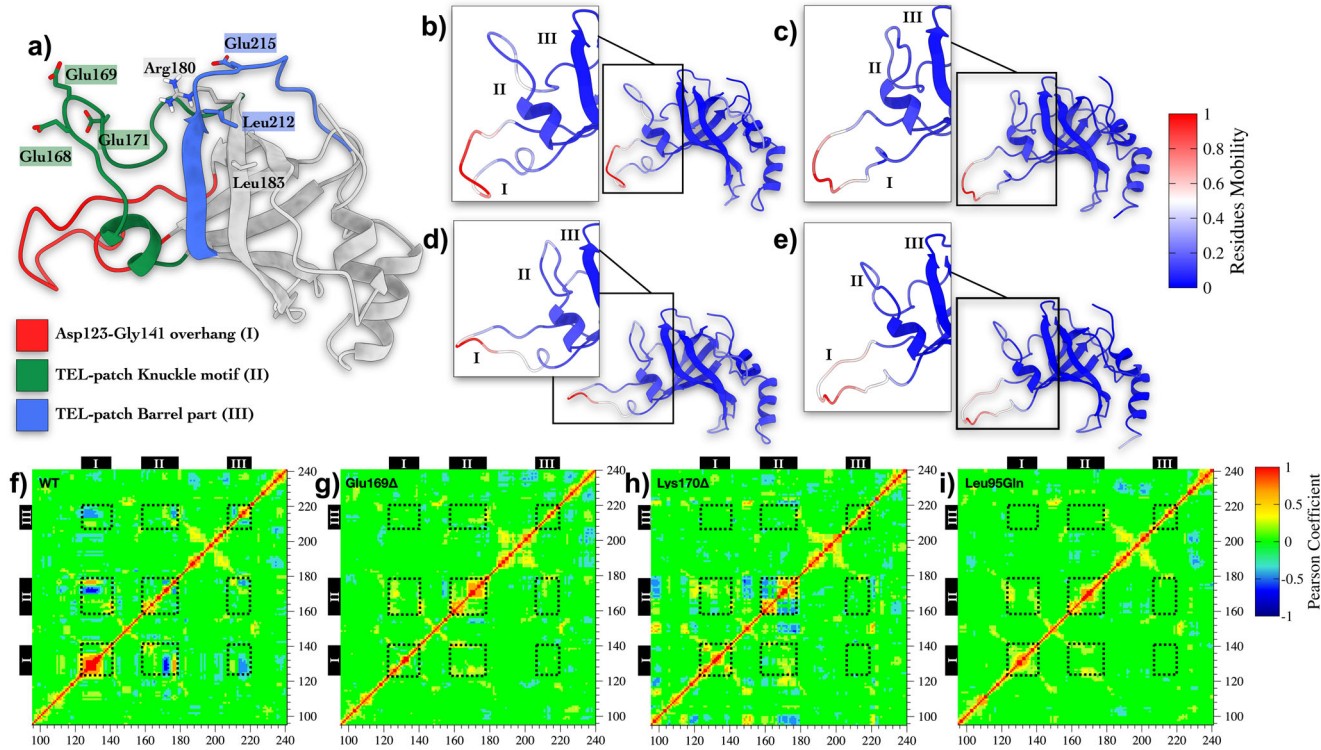

**Fig. 3 Time-series analysis of MD simulations on WT, Glu169Δ, Lys170Δ, and Leu95Gln TPP1. a** Tridimensional representation of TPP1's OB-Domain. **b–e** Projection of modules of the first eigenvector computed for each residue of WT, Glu169Δ, Lys170Δ, and Leu95Gln TPP1, respectively. Large-scale protein motions with slow timescale are depicted in red. **f–i** Pearson coefficients computed between pairs of residues in WT, Glu169Δ, Lys170Δ, and Leu95Gln TPP1 during the MD calculations. Values larger than 0.6 are displayed in red (strong correlation), while those smaller than −0.6 are displayed in blue (strong anti-correlation). Residues forming the three TPP1 regions **I**, **II** and **III** as defined in (**a**) are indicated as black dashed line squares.

**II**, as occurring in Glu169Δ and Lys170Δ variants. The second one is based on the formation of interactions between the N-terminus and region **III**, reducing the conformational flexibility of the latter as seen in the Leu95Gln mutant. While the first mechanism might be easily correlated to the inhibition of TERT processivity (i.e., a TPP1 closed conformation hampers its binding to TERT), the effect of the second one on the TPP1-TERT complex formation is less evident. This motivated us to study the formation and stability of the heterodimeric complex formed by TERT with TPP1 in the WT and variant forms. The outcomes of our investigation are reported in the following section.

**Effect of TPP1 variants on the formation of WT TPP1-TERT complex**. The recently resolved cryo-EM structure of TPP1-TERT heterodimer (PDB ID: 7TRE) prompted us to investigate the impact on the formation of TPP1-TERT binding complex of the TPP1 variants reported in the literature to be associated with pathological phenotypes (see Supplementary Note 5 and Supplementary Table S3). This includes the Glu169Δ, Lys170Δ, and the Leu95Gln variants[28]. The purpose is to provide a molecular understanding of the effect of such variants on the recruitment of TERT by the shelterin protein TPP1. To this end, we performed protein-protein docking calculations between each variant of TPP1 and TERT and analyzed the docking solutions showing the lower RMSD values with respect to the experimental TPP1-TERT structure (PDB ID: 7TRE)[24] (see "Methods" for details). As displayed in Fig. 4a, the deletions Lys170Δ and Glu169Δ affect the folding of the TPP1's α-helix Trp167-Glu171, which is at the binding interface in the TPP1-TERT complex. Consequently, here key inter-protein interactions are lost. In particular, the WT TPP1-TERT complex is characterized by the TPP1's Glu168-

TERT's Arg774 salt bridge, and by the charge-enforced H-bond between TPP1's Glu169 and the side-chain of TERT's Ser134. The deletion of Lys170 causes a partial unfolding of the Trp167-Glu171 sequence, leading to a slightly different TPP1-TERT binding mode (RMSD w.r.t 7TRE ~ 2.6 Å), where TPP1's Glu168 and TERT's Arg774 are no longer able to bind each other. The unfolding of the α-helix Trp167-Glu171 is even more evident in the case of the Glu169Δ variant, where the H-bond between TPP1's Lys170 and TERT's Ser134 is lost (RMSD w.r.t 7TRE ~ 3.1 Å). On the other hand, the mutation Leu95Gln is not at the TEL-patch; however, it affects the correct TPP1-TERT binding mode. In fact, such mutation alters the hydrophobic interactions made by TPP1's Leu95 and Leu181, which favor the formation of the salt bridge between TPP1's Arg92 and TERT's Asp129 (see Fig. 4b). In particular, the Leu95Gln mutation induces the formation of the intra-protein H-bond between Gln95 and Asp148 (see Supplementary Fig. S2), leading to a displacement of TPP1's Arg92 (RMSD w.r.t 7TRE ~ 1.7 Å), with the consequent loss of TPP1's Arg92 and TERT's Asp129 interaction at the binding interface. In order to validate our findings, we performed as blank test a docking calculation between the wild-type forms of TPP1 and TERT, which confirms the experimental binding mode of the complex 7TRE (Fig. 4c). However, one might note that docking calculations have severe limitations in the conformational sampling of binding molecules and large protein conformational changes occurring prior, during or upon the binding, cannot be taken into account by docking calculations that treat proteins as an almost rigid body. To overcome such limitations, we have decided to investigate the structural stability of the TPP1-TERT binding complexes through μs-long MD calculations. Our results show that the WT TPP1 preserves the experimental binding mode as found in PDB ID: 7TRE, with a RMSD value of ~1.5 Å

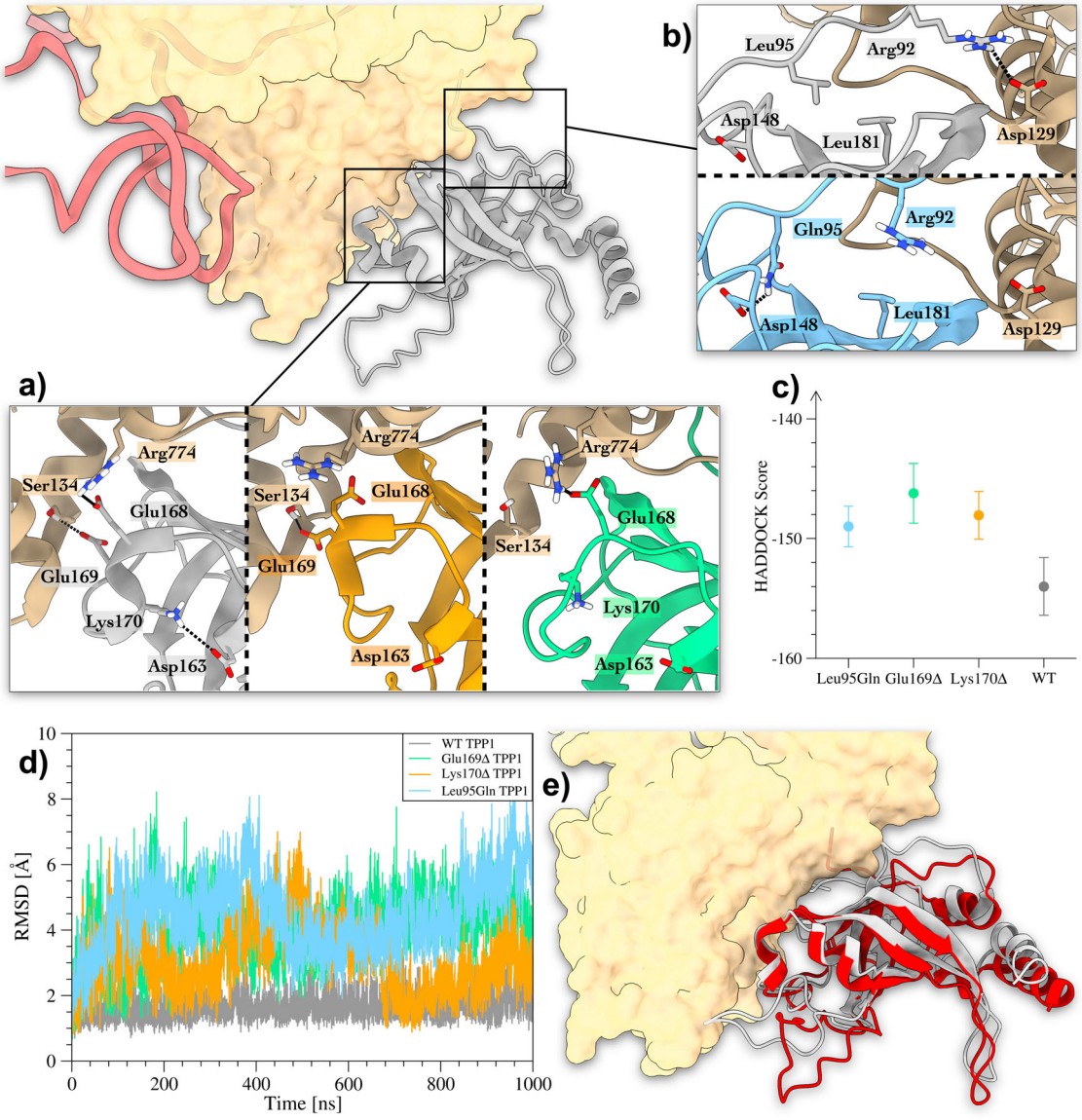

**Fig. 4 Results of docking calculations performed between TERT and TPP1 variants. a** Glu169Δ and Lys170Δ mutants. In the WT TPP1-TERT complex (structure in the left inset), the formation of hTEN's Ser134-TPP1's Glu169 and hTEN's Arg774-TPP1's Glu168 interactions is favored by the intra-protein salt bridge between TPP1's Asp163 and Lys170. When such interactions are lost due to Glu169Δ and Lys170Δ deletions, the TPP1-TERT interaction is weakened (structures in right and central insets, respectively). **b** The Leu95Gln mutant. In the WT TPP1-TERT complex, a salt bridge is formed between hTEN's Asp129 and TPP1's Arg92, favored by the hydrophobic contacts between TPP1's Leu95 and Leu181 (upper inset). The Leu95Gln mutant loses such interaction due to the formation of an interaction between TPP1's Gln95 and Asp148 that distantiates TPP1's Arg92 from hTEN's Asp129 (lower inset). **c** Affinity scores of the best binding modes identified as those with the lower RMSD values with respect to the TPP1-TERT complex reported in PDB ID 7TRE. The standard deviation of each point is represented through the error bar. **d** Plot of the RMSD as a function of simulation time computed for the secondary structure Cα atoms of TPP1 in the WT TPP1-TERT, Glu169Δ TPP1-TERT, Lys170Δ TPP1-TERT, and Leu95Gln TPP1-TERT complexes. WT TPP1 is colored in gray, Glu169Δ in green, Lys170Δ in orange, and Leu95Gln in cyan. **e** Superimposition between the most relevant conformation assumed by WT TPP1-TERT in the MD simulation and the cryo-EM structure 7TRE. The in silico predicted TPP1 pose is shown in gray, while the 7TRE's TPP1 is in red. TERT is colored in beige and represented through its solvent-accessible surface, whereas the TR is shown in magenta.

between the two structures (see Fig. 4e). On the other hand, Glu169Δ TPP1, Lys170Δ TPP1, and Leu95Gln TPP1 lose the experimental binding mode with TERT at the beginning of the simulation. Such behavior is shown by the higher RMSD values computed for the TPP1 backbone as a function of simulation time (Fig. 4d) and the distance values measured between TERT-TPP1 interacting residues (see Supplementary Figs. S9–S13). In particular, the Glu169Δ and Lys170Δ variants are not capable of forming the interactions stabilizing the TPP1-TERT binding such as TPP1's Glu168-TERT's Arg774, TPP1's Glu171-TERT's Ala45,

and TPP1's Thr214-TERT's Asp129. A similar result is observed in the Leu95Gln mutant, though this mutation is far from the TEL-patch (see Supplementary Note 6). In fact, comparing the RMSF profiles between Leu95Gln and WT, one can note the higher values of Leu95Gln at the N-terminus, reflecting an increased mobility for these residues (Supplementary Fig. S9A). In addition, as shown by the protein-protein contact histograms reported in Supplementary Fig. S9B, the Leu95Gln's N-terminus engages in less interactions with TERT compared to the other systems. Upon a closer inspection of the Leu95Gln TPP1-TERT

complex, we found that the N-terminus of this TPP1 variant folds in a short $\alpha$-helix stabilized by a number of intra- and inter-protein H-bonds engaged by Gln95 with TPP1's Gly91, TPP1's Arg92 and TERT's Glu648, which are instead not formed in the TPP1 WT form (see Supplementary Fig. S9C–D). As the final result, key TPP1-TERT interactions like Arg92-Asp129 cannot be established and the binding complex results are unstable. Further details about the MD simulations carried out on the TPP1-TERT heterodimers are reported in the Supplementary Information (see Supplementary Note 7). Overall, our results indicate that the Leu95Gln, Glu169Δ, and Lys170Δ variants significantly affect the binding mode and binding affinity of TPP1 to TERT.

## Conclusion

The molecular binding interaction between the Shelterin protein TPP1 and telomerase is a key player in telomere maintenance mechanism and genome protection. The recently resolved cryo-EM structure of TPP1-telomerase complex (PDB ID 7TRE)[24] has represented a breakthrough in the field, providing the structural basis for understanding the binding interaction between the two proteins. However, experimental structures do not provide information on protein functional dynamics that are relevant for a comprehensive understanding of the molecular recognition process and for rationalizing the effects of pathological variants of the complex. Examples are the deletions Glu169Δ and Lys170Δ—responsible for the telomere-shortening Hoyeraal-Hreidarsson syndrome—and the single-point mutation Leu95Gln—related to Dyskeratosis Cogenita. Atomistic simulations are apt to this scope, being able to complement the experimental structures—obtained by cryo-EM, X-ray or NMR—with the missing conformations for a comprehensive understanding of the protein functional mechanism. Here, we show, by combining microsecond-scale MD calculations, multivariate analysis, cryo-EM, and mutagenesis data, that TPP1's OB-domain is endowed with significant conformational plasticity, regulated by allosteric communications between three regions **I**, **II**, **III**, distant in the protein structure (Asp123-Gly141 overhang, TEL-patch Knuckle motif and TEL-patch barrel part, respectively). Such a communication network allows the TEL-patch region to assume closed and open conformations, functional for the TEL-patch binding to TERT. Interestingly, such a network is lost in the Glu169Δ, Lys170Δ, and Leu95Gln pathological variants of TPP1. In the deletions, the conformational rigidity of the "Knuckle" motif in the TEL-patch (region **II**) does not permit its correct folding into an $\alpha$-helix at the binding interface. On the other hand, the single-point mutation induces the formation of intra- and inter-protein H-bonds formed by Gln95, leading to the folding of the N-terminus into an $\alpha$-helix and a reduced mobility of the OB-domain. As a result, the known landmark interactions formed between TPP1 and TERT are lost, affecting the correct binding with hTEN, which might reduce telomere processivity and cause accelerated aging. Noteworthily, there exists other protein cases where changes in the primary structure induce a different conformational behavior of the protein, with effect on the secondary, tertiary, and even quaternary structure. One relevant example is the p53 protein, a tumor suppressor factor involved in cell growth control and apoptosis activation in case of damage to DNA. Mutation at the level of the core domain, close to the binding site of DNA, may disrupt the conformation of the protein, resulting in loss of function, cell growth, and eventually tumor formation. More specifically, the L3 loop of p53 interacts with the minor groove of DNA, stabilized by a Zn coordination and electrostatic interactions with residue side chains. Upon mutation (e.g., G245S and R249S) these interactions are lost, thus destabilizing the local conformation of the loop and preventing the binding with

DNA.[29] Another interesting example is K-Ras, an important pharmacological target with a high rate of mutations in human cancer. This protein controls signaling networks by switching between active and inactive states with the help of the GTP/GDP cofactor. Specific mutations at the level of the P-loop (i.e., G12X and G13X) connecting the structured regions of $\beta1$ and $\alpha1$, may alter the native tertiary structure by disrupting the interactions between the two regions. This event affects the binding strength of the nucleotide and consequently the biological function of K-Ras.[30] These systems, as TPP1, have an altered conformational flexibility due to punctual modifications in the primary sequence, which perturb their interaction with other partner molecules leading to pathological behavior.

The atomistic structures of wild type and pathological variants of TPP1 in the monomer form and in complex with TERT—released as Supplementary Data files—pave the way to rational drug discovery studies aimed at restoring the correct heterodimer formation. In particular, one could imagine designing allosteric ligands that target the three TPP1 regions and stabilize the open state of the TPP1 OB-domain competent for the recruitment of hTEN. This would restore the normal TPP1/TERT complex and the physiological telomere processivity. Furthermore, the heterodimer complex formed by TPP1 and TERT represents an attractive molecular target to develop anticancer agents. In this case, TEL-patch targeting ligands might inhibit TPP1-telomerase binding and induce cancer cell senescence. Using such an approach, we have recently discovered the first TPP1 ligands with anticancer activity whose pharmacological characterization is underway.

In conclusion, our results endorse atomistic molecular dynamics simulations as a valuable tool to disclose TPP1 functional dynamics, as previously shown in other systems[31–34]. The TPP1-telomerase structures herein presented represent attractive pharmacological targets for drug design of innovative therapeutics in cancer and telomeropathies like Hoyeraal-Hreidarsson syndrome.

## Methods

**MD simulations on WT TPP1, Glu169Δ TPP1, and Lys170Δ TPP1 monomers**. The 3D structure of the TPP1 OB-domain wild type (WT TPP1) was obtained from the protein databank RCSB PDB (PDB ID 2I46)[35]. All the nonprotein species in the crystal structure were removed. The first 5 amino acids (Ser90-Val94) at the N-terminal region were deleted since they are unstructured in the original pdb. The 3D structure of Glu169Δ TPP1 was built with the MODELLER suite[36], removing the Glu169 from the original sequence. The same approach was followed to produce the initial structure of Lys170Δ TPP1. The MODELLER suite was also employed to build the system Leu95Gln TPP1 by replacing Leu95 with Gln95. In all systems, the N-terminal and C-terminal were capped with an acetyl and a methyl-amino protecting group, respectively. Protonation states for amino acids have been assigned using the PropKa algorithm.[37] Then, the proteins were solvated using the TIP3P water model, and a salinity of 0.15 M NaCl. A list of the systems investigated and their composition are provided in Supplementary Note 8 and Supplementary Table S4. The Amber ff99sb-ildn force field was employed[38] with the MD engine GROMACS 2016.5[39]. Each system underwent an energy minimization using the steepest descent algorithm for ~50k steps with a very low force threshold to allow the system to complete all steps. Afterward, the systems underwent a thermalization cycle increasing the temperature from 100 K to 300 K with steps of 50 K, decreasing at each step the restraints applied on the system's heavy atoms. This setting permits solving possible steric clashes between atoms while retaining the overall tertiary structure of the proteins. All systems experienced the following protocol: 200 ps

of NVT simulation followed by 200 ps of NPT simulation for each step of temperature increase. The V-rescale thermostat was employed in the thermalization phase, while the Langevin thermostat in the isothermal-isobaric ensemble was used during the production runs. Periodic boundary conditions have been applied; the short-range electrostatic interactions were computed up to 1 nm, with PME employed for the long-range ones, and van der Waals cutoff set at 1 nm.[40] The pressure was maintained at the reference value of 1 bar using the Parrinello-Rahman barostat[41]. Finally, production runs were run using Langevin MD, with a collision frequency of 10 ps$^{-1}$. The system trajectories were saved every 5000 steps using a time step of 2 fs. The LINCS algorithm was applied to bonds involving hydrogens.

**Docking on TPP1-TERT dimers**. The starting conformations of Glu169Δ, Lys170Δ and Leu95Gln TPP1 were generated through the MODELLER suite, employing TPP1's 7TRE as the initial template[24]. In particular, we employed MODELLER to build the Knuckle motif for Glu169Δ and Lys170Δ, and the N-terminus for the Leu95Gln mutant, keeping the rest of the OB-domain from the TPP1 experimental structure as found in PDB ID 7TRE. Ten models for each TPP1 variant were generated in the absence of TERT. To avoid steric clashes, each model underwent a short energy minimization, where the experimentally derived part of the structure was preserved by applying potential restraints to the positions of the atoms. The 3D coordinates of the complexes formed by Glu169Δ, Lys170Δ and Leu95Gln TPP1 with TERT were obtained by using the molecular docking HADDOCK 2.4 webserver[42]. For docking calculations, we indicated TPP1's Arg92, Glu168, Leu183, Leu212, Pro213, Glu215, and TERT's Lys78, Asp129, Arg132, Leu766, Leu769, Tyr772, Arg774 as points of contact (i.e., active residues), in agreement with the recently released structure of the TPP1-TERT complex (PDB ID 7TRE). The best binding poses among the 200 generated by the docking simulations were selected as those having the lowest RMSD—computed for the secondary structure Cα atoms—with respect to the 7TRE structure.

**MD simulations on WT TPP1-TERT, Glu169Δ TPP1-TERT, Lys170Δ TPP1-TERT, and Leu95Gln heterodimers**. The 3D structures of the WT TPP1-TERT, Leu95Gln TPP1-TERT, Glu169Δ TPP1-TERT, and Lys170Δ TPP1-TERT were obtained from the best results coming from the HADDOCK 2.4 calculations as previously described. Most of the missing residues of the TERT structure in 7TRE (i.e., the moieties Met1-Val10, Asp61-Ser75, Phe101-Pro124, Ala180-Cys321, and Arg416-Thr443) were added using the TET structure with PDB ID: 7QXA[43]. Only the moieties Met1-Arg3, Ala180-Cys321, and Ala418-Thr443 were not modeled. Moreover, in order to preserve the physiological conformational stability of the TERT's hTEN domain, the region Asp105-Pro111 was reconstructed through MODELLER[36]. To ensure structural stability despite the absence of hTR, distance constraints were applied between the IFD and CTD domains through PLUMED 2.8.[44] Regarding the thermalization and production runs, all four TPP1-TERT systems underwent the same protocols employed for the monomers simulations previously described.

**Cluster analysis**. Cluster analyses on the MD trajectories were performed using GROMACS's *gmx cluster* routine, using the *gromos* algorithm. The cluster families of WT TPP1, Glu169Δ TPP1, Lys170Δ TPP1, and Leu95Gln TPP1 were obtained by computing the RMSD for the Cα atoms of the TPP1's residues from Ser165 to Leu184, from Ser210 to Glu215, and the heavy atoms of the side chains belonging to the TEL-patch's amino acids. The RMSD threshold value of 1.8 Å was chosen

considering the number of cluster families generated and the similarity of protein conformations within a cluster family.

**Protein-protein contacts definition**. In order to assess both TPP1's intraprotein interactions and the contacts formed by TPP1 and hTEN at the dimer interface, we computed the frequency of occurrence for each contact, represented as histograms using the *PLOT NA* routine of Drug Discovery Tool (DDT)[45]. We set 4.0 Å as the cutoff distance between two interacting residues.

**Cross-correlation analysis**. Cross-correlation analysis (or Pearson-correlation coefficient analysis) was used to assess the correlated motions between pairs of residues in the monomer and dimer systems. An in-house algorithm was employed to calculate the Pearson coefficients matrix for each couple of residues according to the following formula:

$$C_{ij} = \frac{\langle (r_i - \langle r_i \rangle)(r_j - \langle r_j \rangle) \rangle}{\sqrt{(\langle r_i^2 \rangle - \langle r_i \rangle^2)(\langle r_j^2 \rangle - \langle r_j \rangle^2)}} \quad (1)$$

where $r_i$ and $r_j$ are the position vectors of the Cα atoms in residues $i$ and $j$, respectively. The angle brackets denote time averages computed along the simulations. The final $C_{ij}$ value ranges from $-1.0$ to 1.0. The higher the $C_{ij}$ value, the stronger the linear correlation of the motion of that pair of residues.

**Graph-based network analysis**. The Protein Structure Network (PSN) approach was employed to assess the effect of the Glu169Δ and Lys170Δ mutations on TPP1's network connectivity. Network parameters such as hubs, communities, and structural communication analyses were obtained by using the WebPSN 2.0 webserver[46–48]. The methodology builds the Protein Structure Graph (PSG) based on the interaction strength of two connected nodes:

$$I_{ij} = \frac{n_{ij}}{\sqrt{N_i N_j}} 100 \quad (2)$$

where the interaction percentage ($I_{ij}$) of nodes $i$ and $j$ represents the number of pairs of side-chain atoms within a given cutoff value (4.5 Å), while $N_i$ and $N_j$ are normalization factors. We note that the cutoff values used to describe the interacting residues in DDT and PSN are slightly different since the two methods employ a diverse representation of the system. In DDT, the system is represented all-atom, while in PSN only the heavy atoms of each residue contribute to defining the nodes of the graph. The interaction strength (represented as a percentage) between residues $i$ and $j$ ($I_{ij}$) is calculated for all residue (node) pairs. If $I_{ij}$ is more than the minimum interaction strength cutoff ($I_{min}$) among the residue pairs, then is considered to be interacting and hence represented as a connection in the PSG. The graphs shown in Supplementary Fig. S8 have been generated using the *networkx* library of python3[49].

## Data availability
Atomic coordinates of the TPP1 structures W1, W2, W3, ΔE1, ΔE2, ΔK1, ΔK2, LQ1, and LQ2 are available as Supplementary Data files in the "Supplementary_data_files.zip" archive. A video detailing the effect of Lys170Δ on TPP1 functional dynamics is available at https://youtu.be/AqpRK4659RY and as Supplementary Video 1.

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

## Acknowledgements
This work has received funding from the European Research Council (ERC) under the European Union's Horizon 2020 research and innovation programme ("CoMMBi" ERC grant agreement No. 101001784) and the Swiss National Science Foundation (Project No. 200021_163281). The research was supported by a grant from the Swiss National Supercomputing Centre (CSCS) under project ID s1116 and the Trés Grand Centre de Calcul (TGCC) [project ID ra4681]. The authors thank Robert L. Jernigan and Daniele Di Marino for useful discussions.

## Author contributions
V.L. designed and supervised the research. S.A., V.B.C. and S.R. performed the production and post-processing calculations. All the authors analyzed the results and contributed to the writing of the manuscript.

## Competing interests
The authors declare no competing interests.
