## [Peer Review File · Communications Chemistry]

Conformational plasticity and allosteric communication networks explain Shelterin protein TPP1 binding to human telomeraseReviewers' comments:

Reviewer #1 (Remarks to the Author):

This is a simulation study of a protein called TPP1, which binds the human telomerase enzyme or TERT that is essential for cell lifespan. The authors have compared the wildtype and disease-associated mutant variants using microsecond-long molecular dynamics. The main conclusion is that the wildtype is more flexible, which is essential for its ability to bind TERT. The authors have also characterized the allosteric communication network that connects the binding site with distal sites. The effects of mutations on protein binding are evaluated through protein-protein docking.

Although the results look interesting, there are several issues that need to be fixed before they can be published, which are summarized as follows.

The two mutant variants investigated here are deletions of charged residues in the binding site, which can strongly affect local dynamics and interactions. So, it is not surprising to see some (dynamical) differences between the wildtype and mutants. These differences may simply be a consequence of the mutations, rather than a cause of or directly linked to the loss of binding affinity. More evidence is needed to support the claim that the reduced "conformational freedom" is the basis for the lower binding affinity. A suggestion is to perform a deeper investigation on the L95Q mutation, which is far away from the binding site but still causes weaker binding leading to the disease. The authors only considered this mutation in the docking study, lacking a detailed characterization of the conformational dynamics of the mutant.

Another concern is the use of docking to evaluate binding affinity. The method heavily depends on the starting conformation. The wild-type conformation is directly from the complex structure, so it is ready to bind, whereas for the mutants the conformation may not be so "ready" because they are from modeling (with or without the binding target, which is unclear from the paper). This may generate a bias favoring the wildtype. A fairer comparison is necessary, for example, using the clustering results of MD for the wildtype and mutants.

A lot of technical details are missing, making it difficult to fully understand the work. For example, fix the following issues:

- Was energy minimization performed prior to MD? What parameters were used in EM?
- How frequently was a simulation saved? What was the step size? Any constraints on hydrogen-associated covalent bonds?
- How was the protonation state of ionizable residues determined?
- What collision frequency was used in the Langevin thermostat?
- What clustering algorithm was used? How was the 1.8-angstrom RMSD cutoff determined?
- Was TERT present during the homology modeling of mutants in the docking study? How many models were generated?
- The model having the smallest RMSD to the experimental structure was selected. Was the RMSD based on backbone atoms, C-alpha, or all heavy atoms?

Minor points:

There is a description of contact analysis in the Methods (section 3.3), but there is no result. Different cutoff values (4.0 and 4.5 angstroms) were used for the contact (network) analysis (sections 3.3 and 3.5). Why?

Figure 4 legend. There are some inconsistent descriptions. For example, "(central and right panel, respectively)" does not match the figure. Also, there are two "orange" near the end of the legend. P8, discussion of flexibility/rigidity using PCA eigenvectors. This is inappropriate as eigenvectors are normalized, so not comparable between different systems.

How much percentage of variance is captured for each PC1? A scree plot may be needed for each PCA. Does the docking method consider protein flexibility?

Reviewer #2 (Remarks to the Author):

Simone et. al. presented a technical report on the conformational plasticity and allosteric communication networks explain Shelterin protein TPP1 binding to human telomerase. This manuscript technically sounds interesting presenting computational data to investigate protein-protein interactions, however with lack of experimental validations. This validations by HDX analysis could be a point to consider that could strengthen their hypothesis, however, considering that it may not limit the novelty of this work. Here below are some major concerns;

-This work can be extended in the direction of influence of cancer variants and their influence over the binding of TERT-TPP1 complex.

-Authors used one docking program for protein-protein docking, as a control authors can involve some other docking program.

-From structural point since the binding region is a loop, authors can extend the simulations of the apo TPP1 molecules, and analyze if the loop is stabilized and the secondary structure is defined.

Reviewer #3 (Remarks to the Author):

In the submitted manuscript, Aureli and coworkers investigate three different configurations of the Shelterin complex protein TPP1, which functions as a regulatory protein of the human telomerase TERT. By performing and analyzing molecular simulations, the authors found that significant conformational dynamics exist in the TEL-patch region of the TPP1 wild type, which seems to be important for productive TPP1-TERT binding. This dynamics was severely impaired in the simulated pathological Glu169 Δ and Lys170 Δ variants of TPP1, which are causally associated with diseases such as cancer and Hoyeraal-Hreidarsson syndrome (HHS). The mechanistic conclusions drawn embody the major novelty of the manuscript: a structure-based hypothesis to explanation of this pathological mechanism. The authors also pointed out that the computational results have further applicability in the development of inhibitors that would disrupt the TPP1-TERT protein-protein interaction by binding to the TEL-patch region.

This manuscript provides original mechanistic insights that will be of interest to researchers investigating the basic mechanisms of telomerase regulation and its modulation. It outlines a stimulating hypothesis for further structural or biochemical experiments. The computational methods used are adequate, and sufficient data are provided to allow replication of the performed simulations. Nevertheless, in my opinion, some of the conclusions presented are not fully supported by the current computational results and should be revised.

Main Point:

The claim that "the restricted dynamics of the TEL-patch region in the two pathological variants of TPP1 impedes productive TPP1-TERT binding" is comprehensively examined only on the isolated TPP1 systems. TPP1-TERT protein-protein docking indeed provides some further clues to the complex molecular recognition, but since this manuscript aims to propose a mechanism behind TPP1-TERT complex formation, at least the dynamics and stability of the docking-generated TPP1-TERT key complexes should be investigated in more detail and compared with the WT TPP1-TERT complex. This would require further molecular simulations to be created, performed and evaluated. Especially since no experiments are planned, a more comprehensive investigation of the observed phenomenon would further support the main message of the manuscript.

Further points

In addition, it would be useful to calculate the free energy of binding between each TPP1 variant and TERT for the docked complexes, e.g., using the MM /GBSA method, and possibly even carry out per-residue decomposition. This would add another layer to the understanding of molecular recognition.

Another aspect that should be discussed in more detail is the sampling, as the main manuscript conclusion is based on the observed different dynamics of the TEL-patch region in the pathological variants of TPP1 compared with the broad type. Have you considered creating/simulating any replicates of the TPP1 systems?

Scree plots showing explained variance per each principal component should be generated for all simulations. Preferably, PCA 1 should explain a sufficient amount of variance to link its motion to the proposed biological function.

In addition to the residual mobility shown in Figure 3, porcupine plots of PCA1 showing residual motion should be created and discussed to better show the predominant essential motion through this PC.

The conformational behavior of TPP1 observed in the simulations should be placed in a broader context by comparing it to similar movements/cases already reported in the literature. Is it perhaps possible to at least hypothesize about the broader significance of the observed results?

Please also list the number of atoms of each simulated system in methods section.

REVIEWER 1

Major concerns:

1. *“The two mutant variants investigated here are deletions of charged residues in the binding site, which can strongly affect local dynamics and interactions. So, it is not surprising to see some (dynamical) differences between the wildtype and mutants. These differences may simply be a consequence of the mutations, rather than a cause of or directly linked to the loss of binding affinity. More evidence is needed to support the claim that the reduced “conformational freedom” is the basis for the lower binding affinity. A suggestion is to perform a deeper investigation on the L95Q mutation, which is far away from the binding site but still causes weaker binding leading to the disease. The authors only considered this mutation in the docking study, lacking a detailed characterization of the conformational dynamics of the mutant.”*

We thank the reviewer for raising this point that we have acknowledged performing additional MD calculations on the Leu95Gln mutant. Now all the systems - i.e., wild type, Glu169 Δ , Lys170 Δ , and Leu95Gln - underwent the same type of simulations and the same characterization of the conformational dynamics. The new simulations further confirm that WT TPP1 has a larger conformational flexibility if compared with the variants. Furthermore, we have performed additional μ s-long MD calculations on the complex formed by TERT with TPP1 in the wild type and the variants. Our results show that only WT TPP1 preserves the functional binding mode as experimentally found in PDB ID 7TRE. On the other hand, Glu169 Δ TPP1, Lys170 Δ TPP1, and Leu95Gln TPP1 lose the experimental binding mode with TERT at the beginning of the simulation. Such behavior is shown by the RMSD plots computed for the TPP1 backbone as a function of simulation time (Fig. L1 below and Fig. 4D in the revised manuscript) and the plots of the distances measured between interacting residues in the TERT-TPP1 complex (Fig. S9-S13 in the revised Supplementary Information). In particular, the Leu95Gln mutation affects the TPP1-TERT complex stability due to the formation of intra- and inter-protein H-bonds formed by Gln95, which are instead not formed in the TPP1 WT form. Such H-bonds are engaged with different partners throughout the simulation (i.e., TPP1's Gly91, TPP1's Arg92, and TERT's Glu648) and induces the formation of a short alpha helix at the N-terminus of this TPP1 variant. As final result, key TPP1-TERT interactions like Arg92-Asp129 cannot be formed and the binding complex results unstable.

The results of the additional simulations on Leu95Gln TPP1 mutant and TPP1-TERT complexes are discussed on pages 4-19 and 11 of the revised manuscript, respectively, and in the Supplementary Information.

Fig. L1: Plot of the RMSD as a function of simulation time computed for the secondary structure C α atoms of TPP1 in the WT TPP1-TERT, Glu169 Δ TPP1-TERT, Lys170 Δ TPP1-TERT, and Leu95Gln TPP1-TERT complexes.

2. *“Another concern is the use of docking to evaluate binding affinity. The method heavily depends on the starting conformation. The wild-type conformation is directly from the complex structure, so it is ready to bind, whereas for the mutants the conformation may not be so “ready” because they are from modeling (with or without the binding target, which is unclear from the paper). This may generate a bias favoring the wildtype. A fairer comparison is necessary, for example, using the clustering results of MD for the wildtype and mutants.*

We agree with the Reviewer that docking calculations have severe limitations in the conformational sampling of binding molecules, and large protein conformational changes occurring prior, during or upon the binding, cannot be taken into account by docking calculations that treat proteins as an almost rigid body. To overcome docking limitations, we have decided to investigate the binding complexes formed by TERT and TPP1 in its wild-type and variant forms by means of μ s-long MD calculations, where the conformational dynamics and the stability of the protein-protein binding mode was characterised (please see previous point 1).

3. *“A lot of technical details are missing. Was energy minimization performed prior to MD? What parameters were used in EM? How frequently was a simulation saved? What was the step size? Any constraints on hydrogen-associated covalent bonds? How was the protonation state of ionizable residues determined? What collision frequency was used in the Langevin thermostat? What clustering algorithm was used? How was the 1.8-angstrom RMSD cutoff determined? Was TERT present during the homology modeling of mutants in the docking study? How many models were generated? The model having the smallest RMSD to the experimental structure was selected. Was the RMSD based on backbone atoms, C-alpha, or all heavy atoms?”*

We apologize for the lack of such information in the original text. We have now added

all the requested technical details in the revised “Methods” section and reported them below for the Reviewer convenience.

- Energy minimization was performed using the steepest descent algorithm for ~50k steps with a very low force threshold to allow the system to complete all steps. Short range electrostatic interactions were computed up to 1 nm, with PME employed for the long-range ones, and van der Waals cutoff set at 1 nm;
- The thermalization was carried out as reported in the manuscript, using the same nonbonded parameters, alternating cycles of NPT and NVT simulations in order to equilibrate the system and achieve the best conditions for the production run;
- During the production runs, the system trajectories were saved every 5000 steps using a time step of 2 fs. The LINCS algorithm was applied to bonds involving hydrogens, and Langevin collision frequency was set to 10 ps⁻¹.
- The protonation state of each residue was assigned using propka.
- As for clusterization, the algorithm used was “Gromos” and the RMSD threshold value of 1.8 Å was chosen considering the number of cluster families generated and the similarity of protein conformations within a cluster family.
- Regarding the modeling of TPP1 variants, we employed MODELLER to build the “Knuckle motif” for Glu179Δ and Lys170Δ, and the N-terminus for Leu95Gln mutant, keeping the rest of the OB-domain from the TPP1 experimental structure in complex with TERT as found in PDB ID 7TRE. Ten models for each TPP1 variant were generated in absence of TERT. To avoid steric clashes, each model underwent a short energy minimization, where the experimentally derived part of the structure was preserved by applying potential restraints to the positions of the atoms.
- The RMSD between the 200 complex models obtained from docking and the experimental structure was computed for the secondary structure C α atoms.

Minor points

4. *“There is a description of contact analysis in the Methods (section 3.3), but there is no result.”*

We thank the Reviewer for pointing this out. In the original manuscript, we employed the contact analysis to generate Fig. S1. In the revised manuscript, we used the same analysis to elucidate the contacts between TERT and TPP1 in the wild type and the variants. Following the Reviewer’s suggestion, we refer to the Methods section when this analysis is discussed in the revised text.

5. *“Different cutoff values (4.0 and 4.5 angstroms) were used for the contact (network) analysis (sections 3.3 and 3.5). Why?”*

The cut-off values used to describe the interacting residues in DDT (contact) and PSN (network) are slightly different since the two methods employ a diverse representation of the system. In DDT, the system is represented all-atom, while in PSN the system is represented as graph. We further note that the PSN cutoff of 4.5 Å is the default value and to the best of our knowledge, it cannot be modified.

We have reported this difference in the revised manuscript at page 15.

6. *“Figure 4 legend. There are some inconsistent descriptions. For example, “(central and right panel, respectively)” does not match the figure. Also, there are two “orange” near the end of the legend.”*

We thank the reviewer for the comment and we have improved the clarity of Figure 4 legend.

7. *“P8, discussion of flexibility/rigidity using PCA eigenvectors. This is inappropriate as eigenvectors are normalized, so not comparable between different systems. How much percentage of variance is captured for each PC1? A scree plot may be needed for each PCA.”*

We have acknowledged the Reviewer’s comment reporting below and on page 5 of the revised Supplementary Information the percentage of explained variance for PC1 and the scree plots for each replica of the investigated systems (see Tab. L1 and Fig. L2 below).

Furthermore, prompted by the Reviewer’s comment, in the revised manuscript we have also reported the porcupine plots of the first two eigenvectors, the normalized projection of the second eigenvalue, and the scree plot for each replica (please see Fig. S3, S4, S5, and S6 in the revised Supplementary Information).

System	Replica 1 (r1)	Replica 2 (r2)	Replica 3 (r3)
WT	50% (PC2: 14%)	57% (PC2: 10%)	34% (PC2: 14%)
E169Δ	55% (PC2: 11%)	28% (PC2: 17%)	30% (PC2: 14%)
K170Δ	29% (PC2: 15%)	37% (PC2: 14%)	42% (PC2: 17%)
L95Q	30% (PC2: 18%)	41% (PC2: 12%)	68% (PC2: 5%)

Tab. L1: Percentage of explained variance calculated for the first principal component PC1 in each simulation replica for the systems WT TPP1, Glu169Δ TPP1, Lys170Δ TPP1, and Leu95Gln TPP1. Furthermore, the percentage of explained variance calculated for the second component PC2 is reported in parenthesis.

Fig. L2: Results of the PCA carried out on the MD simulations performed on WT TPP1 and its variants. A-D) Scree plots showing the percentage of explained variance for each component. The outcomes of WT TPP1 are colored in grey, Glu169Δ TPP1 in green, Lys170Δ TPP1 in orange, and Leu95Gln in cyan.

8. “Does the docking method consider protein flexibility?”

In our docking study we employed the software HADDOCK that performs conformational sampling of the residue side chains at the protein-protein binding interface, while protein backbone is rigid. The MD calculations performed on the TPP1-TERT complexes allow overcoming the sampling limitation of docking since both backbone and side chains atoms are fully flexible (see previous point 1).

REVIEWER 2

1. “This work can be extended in the direction of influence of cancer variants and their influence over the binding of TERT-TPP1 complex.”

We thank the Reviewer for this comment that we have acknowledged performing a total of 4 μ s of additional MD calculations on the complex formed by TERT with TPP1 in the wild type and the variants associated to pathologies as reported in Tab S3. Our results show that only WT TPP1 preserves the functional binding mode as experimentally found in PDB ID 7TRE. On the other hand, Glu169Δ TPP1, Lys170Δ TPP1, and Leu95Gln TPP1 lose the experimental binding mode with TERT at the beginning of the simulation. Such behavior is shown by the RMSD plots computed for the TPP1 backbone as a function of simulation time (Fig. 4D in the revised manuscript) and the plots of the distances measured between interacting residues in the TERT-TPP1 complex (Fig. S9-S13 in the revised Supplementary Information). The results of the additional simulations are discussed on page 11 of the revised text. For the Reviewer convenience, we report below the RMSD plots of the diverse variants of TPP1 in complex with TERT.

Fig. L3: Plot of the RMSD as a function of time computed for the secondary structure C α atoms of TPP1 in the WT TPP1-TERT, Glu169 Δ TPP1-TERT, Lys170 Δ TPP1-TERT, and Leu95Gln TPP1-TERT complexes.

2. “Authors used one docking program for protein-protein docking, as a control authors can involve some other docking program.”

We have acknowledged the Reviewer’s comment performing additional docking calculations by means of ClusPro, an ab-initio docking software that uses the FFT-based Piper algorithm (doi: 10.1038/nprot.2016.169). The new docking simulations show that WT TPP1 has the lowest score, confirming the results reported in the original manuscript using the HADDOCK docking program as displayed in the figure below. Furthermore, we note that in the revised manuscript the complexes formed by TERT with TPP1 in the wild type and the variants have been further investigated by μ s-long MD calculations, which allowed assessing the structural stability of the binding complexes considering the full flexibility of proteins’ backbone and side chains, a more physiological condition than docking.

Fig. L4: Affinity scores of the best binding modes identified as those with the lower RMSD values with respect to the experimental TPP1-TERT complex with PDB ID 7TRE. On the left, the results obtained using the HADDOCK program, while on the right, the results obtained by means of ClusPro.

3. “From structural point since the binding region is a loop, authors can extend the simulations of the apo TPP1 molecules, and analyze if the loop is stabilized and the secondary structure is defined.”

Following the Reviewer’s suggestion, we have extended the WT TPP1 simulation to 3 μ s, three times the simulation time reported in the original manuscript. The results are displayed in the figures below that show the agreement of the data collected from the additional calculations with those reported in the original manuscript in Fig. 2. Furthermore, in the revised manuscript we report two replicas for each system - i.e., wild-type and variant forms - for a total of 12 μ s of calculations. The obtained results are shown in Fig. 2 and discussed in the main text and Supplementary Information.

Fig. L5: Comparison of the results obtained from the original 1 μ s MD simulation (grey lines) and the extended 3 μ s simulation (red lines) on WT TPP1. A) Plot of the RMSD as a function of simulation time computed for the secondary structure Ca atoms of TPP1. B) Plot of RMSF values computed for each residue of WT TPP1.

REVIEWER 3

Main Point:

1. “The claim that “the restricted dynamics of the TEL-patch region in the two pathological variants of TPP1 impedes productive TPP1-TERT binding” is comprehensively examined only on the isolated TPP1 systems. TPP1-TERT protein-protein docking indeed provides some further clues to the complex molecular recognition, but since this manuscript aims to propose a mechanism behind TPP1-TERT complex formation, at least the dynamics and stability of the docking-generated TPP1-TERT key complexes should be investigated in more detail and compared with the WT TPP1-TERT complex. This would require further molecular simulations to be created, performed and evaluated. Especially since no experiments are planned, a more comprehensive investigation of the observed phenomenon would further support the main message of the manuscript.”

We fully agree with the Reviewer and we have acknowledged the comment performing μ s-long MD calculations on the TPP1-TERT complexes for the wild type and pathological variants. The results show that only the wild-type complex is able to maintain the functional binding complex as experimentally found in PDB ID 7TRE. Instead, in all the TPP1 variants, the experimental binding mode between TPP1 and TERT is lost at the beginning of the simulation. Such behavior is shown by the RMSD plots computed for the TPP1 backbone as a function of simulation time (please see Fig. L6 at the following point 2 or Fig. 4D in the revised manuscript) and the plots of

the distances measured between interacting residues in the TERT-TPP1 complex (Fig. S9-S13 in the revised Supplementary Information). In particular, the Glu169 Δ and Lys170 Δ variants - characterized by a deletion of a key amino acid at the TEL-patch's Knuckle motif - are not capable of forming the interactions stabilizing the TPP1-TERT binding (e.g., TPP1's Glu168-TERT's Arg774, TPP1's Glu171-TERT's Ala45, TPP1's Thr214-TERT's Asp129). A similar result is observed in the TPP1 Leu95Gln mutant, though this mutation is far from the TEL-patch. In such a case, the Leu95Gln mutation affects the TPP1-TERT complex stability due to the formation of intra- and inter-protein H-bonds engaged by Gln95, which are instead not formed in the WT TPP1 form. Such H-bonds are engaged with different partners throughout the simulation (i.e., TPP1's Gly91, TPP1's Arg92, and TERT's Glu648) and induces the formation of a short alpha helix at the N-terminus of this TPP1 variant. As final result, key TPP1-TERT interactions like Arg92-Asp129 cannot be formed and the binding complex results unstable. The results of the additional simulations on TPP1-TERT complexes are discussed on page 11 of the revised manuscript and in the Supplementary Information. These simulations add an important piece of information that strengthens the message of our work. We especially thank the Reviewer for this suggestion.

Further points:

2. *"In addition, it would be useful to calculate the free energy of binding between each TPP1 variant and TERT for the docked complexes, e.g., using the MM /GBSA method, and possibly even carry out per-residue decomposition."*

Prompted by the Reviewer's comment at previous point 1, we have performed MD calculations on the complexes formed by TERT with TPP1 in its wild type and variant forms. During these simulations, the binding mode between TERT and TPP1 is stable only for the wild type, while in the three variants (Glu169 Δ , Lys170 Δ , and Leu95Gln) the TERT-TPP1 interactions are broken, and the experimental binding mode is lost. Such results indicate that the wild type is the energetically most stable complex, therefore, performing free-energy calculations like MM/GBSA are not necessary. The different stability of the diverse complexes is shown by the figure below - also Fig. 4D in the revised manuscript - reporting the RMSD values computed as a function of time for the secondary structure C α atoms of TPP1 at the binding interface with TERT for each system.

Fig. L6: Plot of the RMSD as a function of simulation time computed for the secondary structure C α atoms of TPP1 in the WT TPP1-TERT, Glu169 Δ TPP1-TERT, Lys170 Δ TPP1-TERT, and Leu95Gln TPP1-TERT complexes.

3. *“Another aspect that should be discussed in more detail is the sampling, as the main manuscript conclusion is based on the observed different dynamics of the TEL-patch region in the pathological variants of TPP1 compared with the broad type. Have you considered creating/simulating any replicates of the TPP1 systems?”*

We have acknowledged the Reviewer’s comment by performing additional MD calculations for all the four variants of TPP1 (WT, Glu169 Δ , Lys170 Δ and Leu95Gln). In particular, we have carried out three replicas for each system (12 in total). The number of conformations - and consequently cluster families - sampled by TPP1 increased, as expected. However, the conformational behavior reported in the original manuscript was confirmed with WT TPP1 showing the larger conformational changes if compared with the other systems. The results of the additional calculations are reported and discussed on pages 4-7 and in Figure 2 of the revised manuscript.

4. *“Scree plots showing explained variance per each principal component should be generated for all simulations. Preferably, PCA 1 should explain a sufficient amount of variance to Link its motion to the proposed biological function.”*

As requested by the Reviewer, we have added the scree plots computed for each system in the revised Supporting Information and reported below for the Reviewer convenience. As can be seen, PCA 1 has in all cases the highest percentage of explained variance indicating that the associated protein motion represents a functionally relevant slow degree of freedom of the system.

Furthermore, prompted by the Reviewer’s comment, in the revised manuscript we have also reported the porcupine plots of the first two eigenvectors and the normalized projection of the second eigenvalue, together with the scree plot for each replica (please see Fig. S3, S4, S5, and S6 in the revised Supplementary Information).

Fig. L7: Results of the PCA carried out on the MD simulations performed on WT TPP1 and its variants. A-D) Scree plots showing the percentage of explained variance computed for each component. The outcomes of WT TPP1 are colored in grey, Glu169 Δ TPP1 in green, Lys170 Δ TPP1 in orange, and Leu95Gln TPP1 in cyan.

5. *“In addition to the residual mobility shown in Figure 3, porcupine plots of PCA1 showing residual motion should be created and discussed to better show the predominant essential motion through this PC.”*

We have fulfilled the Reviewer's request reporting in the revised Supplementary Information the porcupine plots of the first eigenvector for WT and TPP1 variants. In addition, we have also computed the porcupine plots of the second eigenvector for all the systems (please see Fig. S3, S4, S5, and S6 in the revised Supplementary Information). These plots together with the scree plots showing the percentage of explained variance computed for each principal component, indicate that PCA 1 is the predominant essential motion of the system.

6. *“The conformational behavior of TPP1 observed in the simulations should be placed in a broader context by comparing it to similar movements/cases already reported in the literature. Is it perhaps possible to at least hypothesize about the broader significance of the observed results?”*

We thank the Reviewer for raising this interesting point. Indeed, there are other protein cases where changes in the primary structure induce a different conformational behavior of the protein, with effect on the secondary, tertiary and even quaternary structure.

One relevant example is the p53 protein, a tumor suppressor factor involved in cell growth control and apoptosis activation in case of damage to DNA. Mutation at the level of the core domain, close to the binding site of DNA, may disrupt the conformation of the protein, resulting in loss of function, cell growth and eventually tumor formation. More specifically, the L3 loop of p53 interacts with the minor groove of DNA, stabilized by a Zn coordination and electrostatic interactions with residue side chains. Upon mutation (e.g., G245S and R249S) these interactions are lost, thus destabilizing the

local conformation of the loop and preventing the binding with DNA as reported in doi: 10.1073/pnas.96.15.8438.

Another interesting example is K-Ras, an important pharmacological target with a high rate of mutations in human cancer. This protein controls signaling networks by switching between active and inactive states with the help of the GTP/GDP cofactor. Specific mutations at the level of the P-loop (i.e., G12X and G13X) connecting the structured regions of $\beta 1$ and $\alpha 1$, may alter the native tertiary structure by disrupting the interactions between two regions. This event affects the binding strength for the nucleotide and consequently the biological function of K-Ras (doi: 10.1016/j.csbj.2020.04.003).

These systems, as TPP1, have an altered conformational flexibility due to punctual modifications in the primary sequence, which perturb their interaction with another partner molecules leading to pathological behavior.

These examples are now reported in the revised version of the manuscript on page 15.

7. *“Please also list the number of atoms of each simulated system in methods section.”*

Following the Reviewer’s request, we have prepared the table below reporting the number of atoms for each system investigated. The table is also added to the Supplementary Information as Tab. S4.

System	Number of atoms*
WT-TPP1 (monomer in solution)	41320
E169 Δ -TPP1 (monomer in solution)	50999
K170 Δ -TPP1 (monomer in solution)	48827
L95Q-TPP1 (monomer in solution)	50980
WT-TPP1-TERT Complex	284904
E169 Δ -TPP1-TERT Complex	280705
K170 Δ -TPP1-TERT Complex	286498
L95Q-TPP1-TERT Complex	298887

**Total number of atoms including water molecules and salt ions (NaCl)*

REVIEWERS' COMMENTS:

Reviewer #1 (Remarks to the Author):

The authors have done excellent work to address all my points. The only (minor) thing I want to mention is the writing "In DDT, the system is represented all-atom, while in PSN the system is represented as a graph." I believe PSN uses all HEAVY (non-hydrogen) atoms, which may be different from DDT. Otherwise, the paper is well-revised and should be accepted for publication.

As requested by the editor, I also checked the authors' responses to the questions by Reviewer #2. In my opinion, the authors have addressed all the previous concerns of the reviewer.

Reviewer #3 (Remarks to the Author):

In the submitted revised version of the manuscript the authors have adequately and diligently addressed all the comments and concerns raised and substantially improved the quality and scope of the results presented.

As a result, I recommend the acceptance of the revised manuscript for publication in its current form.

Reviewer 1

1) “The authors have done excellent work to address all my points. The only (minor) thing I want to mention is the writing “In DDT, the system is represented all-atom, while in PSN the system is represented as a graph.” I believe PSN uses all HEAVY (non-hydrogen) atoms, which may be different from DDT. Otherwise, the paper is well-revised and should be accepted for publication.”

Following the suggestion of the Reviewer, we have revised the sentence as follows:

“In DDT, the system is represented all-atom, while in PSN the system is represented as a graph where nodes correspond to the heavy atoms of the protein.”